# Expression of Hormones, Cytokines, and Antioxidants in Heat-Stressed Subfertile Female Dromedaries

**DOI:** 10.3390/ani12162125

**Published:** 2022-08-19

**Authors:** Moustafa M. Zeitoun, Derar R. Derar, Ahmed Ali, Yousef M. Alharbi

**Affiliations:** 1Department of Animal Production and Breeding, College of Agriculture and Veterinary Medicine, Qassim University, Buraydah 52571, Saudi Arabia; 2Department of Animal and Fish Production, Faculty of Agriculture, El-Shatby, Alexandria University, Alexandria 21545, Egypt; 3Department of Veterinary Medicine, Qassim University, Buraydah 52571, Saudi Arabia; 4Department of Theriogenology, Faculty of Veterinary Medicine, Assiut University, Assiut 71515, Egypt

**Keywords:** camel, infertility, heat stress, hormones, antioxidants

## Abstract

**Simple Summary:**

Heat stress imposes a high burden on domestic animals’ productive and reproductive performance. Due to the long hot summer, drought, and shortage of green fodders, camels raised in the desert suffer a lot of reproductive inefficiencies. This animal represents one of the main wealth sources for the desert inhabitants. Several fertility disorders have been discovered, leading to frequent breeding without pregnancy. This study aimed at exploring blood metabolites such as metabolic and reproductive hormones, cytokines, and antioxidants to be monitored as bio-indictors for subfertility in female camels. The results confirmed that none of the tested metabolic hormones and glucose revealed differences among fertile and subfertile females. However, FSH, inhibin, IL-ß, nitrous oxide, and glutathione revealed remarkable differences between fertile and subfertile females, which would be reliable tools to predict subfertility statuses in this animal. IL-ß revealed higher levels in the cases with genital inflammations. The normal profiles in control females revealed the highest FSH, and the lowest inhibin were vice versa in all subfertile females. Nonetheless, nitrous oxide and glutathione would also be reliable bio-indicators for judging the fertility status.

**Abstract:**

The prevailing hot climate imposes heavy burdens on the productivity of the camel, goat, and sheep herds raised in the Gulf desert. Due to the lack of a reliable indicator for the various subfertility statuses in camel females, this study aimed to investigate the expression of inhibin, TGFά, ILß, FSH, sex and metabolic hormones, and antioxidants for the fertility status in camel females. Eighty-two subfertile and five fertile females were admitted to the university clinic with the complaint of repeat breeding with failed conception. The animal’s genital tracts were examined for reproductive soundness. Blood samples were withdrawn for hormonal, cytokines, and antioxidants determinations. Subfertile females were categorized into six groups; endometritis (EN, 28), inactive ovaries (IO, 20), ovarian hydrobursitis (BU, 19), vaginal adhesions (VA, 7), salpingitis (SA, 4), and cervicitis (CE, 4). Results revealed a significant increase in inhibin in all groups compared to control (68.2, 66.4, 61.8. 58.8, 58.3, 55.8, and 36 pg/mL, in CE, VA, IO, BU, EN, SA, and CON, respectively). TGFά, dehydroepiandrosterone (DHEA), and progesterone were not different among groups, whereas IL-ß differed among groups. FSH, estradiol, nitrous oxide, and glutathione were higher in CON compared with other groups. In conclusion, reproductive failures in camel females are reflected in the imbalances of endocrine, cytokines, and antioxidants bio-indicators.

## 1. Introduction

The harsh desert climate has been known for decades to impair reproduction in mammals. Several organism responses might occur in the body, reflecting its resistance to the adverse surroundings. These responses are: immunosuppression, fatigue, diseases, low production, and low fertility or even sterility. For decades, the research on animal subfertility has focused on the roles played by the reproductive hormones and the endocrinological imbalances. However, recently, the ongoing research targeted the roles of the small peptides, i.e., cytokines and the enzymatic and non-enzymatic antioxidants on the reproductive process. A study has been conducted on the effects of various cytokines on the development and regulation of the hypothalamic–hypophyseal–gonadal axis, ovarian follicular growth, implantation, and immune system modulation during pregnancy in rats [1]. Ingman and Jones revealed imbalances in both pituitary gonadotropins and gonadal steroids [1]. Cytokines are produced by several types of cells, including endometria causing endometriosis-associated subfertility in case of their elevations [2]. The family of Interleukins (IL), TNFά, IFN-γ, and monocyte chemo-attractant protein-1 (MCP-1) were highly increased in the peritoneal fluid of the women suffering from subfertility and chronic infectious inflammation of the reproductive system [3]. Inflammation of the genital tract elicits leukocytes’ entrance, which secretes various types of cytokines and leads to implantation failure [4]. Recently, Poole and his colleagues found a significant increase in IL6 in non-pregnant versus pregnant synchronized postpartum cows’ uterine flushing [5]. However, they found the opposite trend for TGFß, which revealed increases in pregnant versus non-pregnant cows.

Meanwhile, inhibin has long been known for its double actions on folliculogenesis inhibition and reducing FSH secretion from the anterior pituitary [6]. Sharma and his coworkers investigated the in vitro effects of several growth factors, including IGF-I, TGFά, TGFß1, and bFGF, used alone or in combination with FSH on the development, survival, antrum, and apoptosis of buffalo preantral follicles [7]. TGFά and TGFß inhibited follicular survival and increased the incidence of oocyte death, while the combination of IGF-I, TGFά, and TGFß1 alleviated the negative effect of the TGFs. Moreover, interleukin (IL)-1 has been known to be a key player in reestablishing some events in the seminiferous epithelium of the testis during spermatogenesis [8]. Tibary and Anouassi reported that the Arabian one-humped female camels suffer several reproductive inefficiencies such as poor estrous display, short breeding season, and high incidence of early embryonic deaths, repeat breeding without conception, low conception rate, and extended calving interval [9]. Therefore, the main goal of the present study was to elucidate the association between the blood levels of the ovarian steroids, metabolic hormones, cytokines, and antioxidants to the common subfertility inefficiencies in female dromedaries inhabiting the desert. Moreover, the study aimed to find out a strong bio-indicator in subfertile female camels exposed to heat stress.

## 2. Material and Methods

### 2.1. Animals and Location

Eighty-two female dromedary camels were admitted to the veterinary teaching clinic at Qassim University, the central region of Saudi Arabia, from February to July 2020 for breeding soundness examinations, and the owners’ complaints were recorded. Additionally, five normal cyclic fertile females were also investigated and included. The reproductive tract of each animal was examined using standard trans-rectal palpation and by ultrasonography attached to a 5 MHz probe (Aloka SSD 500; Aloka Co., Ltd., Tokyo, Japan). The vagina was examined manually with a gloved hand to estimate the patency of the vagina and cervix and to evaluate the nature of the vaginal discharges. All animals (including the control) were bled once at the time of diagnosis by jugular venipuncture into non-heparinized Vacutainer^®^ tubes. Endometritis, inactive ovaries, ovarian hydrobursitis, vaginal adhesions, salpingitis, and cervicitis were diagnosed in 28 (34.14%), 20 (24.39%), 19 (23.17%), 7 (8.54%), 4 (4.87%), and 4 (4.87%) of the barren females, respectively.

### 2.2. FSH Determination

Camel FSH was determined by a commercial Sandwich ELISA kit (MyBioSource, San Diego, CA, USA) according to Rebar et al. [10]. The assay sensitivity was 0.1 mIU/mL. The detection range of the assay is 0.625–20 mIU/mL; however, the intra-assay CV was 10.7%.

### 2.3. Estradiol-17ß Determination

Estradiol was determined according to Ratcliffe et al. [11] using a commercial competitive ELISA kit (Human Gesellschaft für Biochemica und Diagnostica mbH, Wiesbaden, Germany). The assay’s sensitivity was 13 pg/mL, the detection range was 25–2000 pg/mL, and the intra-assay CV was 8.4%.

### 2.4. Progesterone Determination

A commercial competitive ELISA kit determined progesterone concentrations in camel sera (Human Gesellschaft für Biochemica und Diagnostica mbH, Wiesbaden, Germany) according to Rawanska et al. [12]. The assay’s sensitivity was 0.03–0.07 ng/mL, the detection range was 0.3–40 ng/mL, and intra- and inter-assay CV were 6.5 and 9.7%, respectively.

### 2.5. Camel Inhibin Determination

Inhibin concentrations were determined according to Kricka [13] using a commercial ELISA kit specific for camels (MyBioSource, San Diego, CA, USA). The sensitivity was 2 pg/mL, the detection range was 16.5–500 pg/mL, and the intra-assay CV was less than 15%.

### 2.6. DHEA Determination

Dehydroepiandrosterone (DHEA) concentrations were determined by a commercial simple solid-phase ELISA (DIAsource ImmunoAssays, Louvain-la-Neuve, Belgium). The sensitivity is 82 pg/mL, the intra-assay CV is 9.9%, and the lower detection limit is 0.2 ng/mL. The determination was conducted according to the method previously mentioned [14].

### 2.7. Camel TGFά Determination

Camel TGFά determination was performed according to Mouradian et al. [15] using a commercial sandwich ELISA kit (MyBioSource, San Diego, CA, USA). The assay’s sensitivity was 1 pg/mL, the detection range was 3.12–100 pg/mL, and the intra-assay CV was 13%.

### 2.8. ILß Determination

The camel interleukin ß concentrations were determined according to Herzyk and Wewerk [16] by a commercial sandwich ELISA (Elabscience, Huston, TX, USA). The assay’s sensitivity was 4.69 pg/mL, the detection range was 7.81–500 pg/mL, and intra-assay C.V. was 9.8%.

### 2.9. GSH Determination

According to Meister and Anderson, reduced glutathione concentrations were determined based on its reaction with Dinitrobenzoic acid (DNTB, Elabscience, Huston, TX, USA), forming yellow color measured at 405 nm [17]. The assay’s sensitivity was 2 µmol/L, and the detection range was 2–100 µmol/L.

### 2.10. NO Determination

Nitrous oxide was determined by a commercial kit (Elabscience, Huston, TX, USA) according to the method stated above [18]. The method is based on NO oxidation to form NO2, which reacts with an azo compound, forming a color whose intensity is proportional to the concentration of NO in serum. The assay’s sensitivity was 0.16 µmol/L, and the detection range was 0.16–100 µmol/L.

### 2.11. Free T4 Determination

Free thyroxine (FT4) was determined by a commercial kit (Human Gesellschaft für Biochemica und Diagnostica mbH, Wiesbaden, Germany) according to the method of Sterling [19]. The assay’s sensitivity was 0.06 ng/mL, the detection range was 0.12–8 ng/mL, and the intra- and inter-assay CV were 3.9 and 5.7%, respectively.

### 2.12. Free T3 Determination

Free thyroxine (FT3) was determined by a commercial kit (Human Gesellschaft für Biochemica und Diagnostica mbH, Wiesbaden, Germany) according to the method previously mentioned [20]. The assay’s sensitivity was 0.2 ng/mL, the detection range was 0.2–10 ng/mL, and the inta- and inter-assay CV were 5.6 and 7.9%, respectively.

### 2.13. Total T4 Determination

Total thyroxine (T4) was determined by a commercial kit (Human Gesellschaft für Biochemica und Diagnostica mbH, Wiesbaden, Germany) according to the method of Sterling [19]. The assay’s sensitivity was 0.06 ng/mL, the detection range was 2–25 µg/dL, and intra- and inter-assay CV were 3.9 and 5.7%, respectively.

### 2.14. Statistical Analysis

Data of hormones, cytokines, and antioxidants were analyzed using one-way ANOVA of the general linear model by SAS software version 9.0 [21]. The statistical model is described as follows:Y_ij_ = µ + Ai + *eij*

where; Y_ij_ is the observation taken on the jth individual,

µ = overall mean,Ai = the fixed effect of the ith fertility status, (i = 1……7).*eij* = random error assumed to be independent normally distributed with mean = 0 and variance = Ợ^2^.

The Duncan’s Multiple Range Test (DMRT) conducted mean comparisons among groups, and the significance level was considered at *p* < 0.05. Pearson correlations among hormones, cytokines, and antioxidants were estimated within groups [21].

## 3. Results

Data in Table 1 demonstrate that FSH was highest (*p* < 0.05) in the fertile (CON) than in the infertile females. The second highest levels of FSH were found in females with cervicitis (CE) and vaginitis (VA). Inhibin, however, exhibited an opposite trend to FSH, showing its lowest level in the fertile control females but higher (*p* = 0.0005) levels in all groups of subfertility. The levels of inhibin approached about two-fold in the cases of cervicitis and vaginitis compared with that in control. Estradiol 17ß revealed the highest levels in control and females suffering from Fallopian tube adhesions. However, the other categories of subfertility have demonstrated fewer levels of estradiol. Contrariwise, there were no significant (*p* > 0.05) changes in progesterone levels among groups. Likewise, there were non-significant differences among groups in DHEA concentrations, but a trend of elevated DHEA levels associated with the cases having inflammations (i.e., uterine, cervical, and vaginal).

Table 2 illustrates the data of glucose and metabolic hormones in fertile and subfertile female camels. Results revealed non-significant (*p* > 0.05) differences among groups in plasma concentrations of total T4, free T4, free T3, and glucose, even though there was a trend of glucose elevation in the fertile compared with subfertile females.

Table 3 exhibits the data of the cytokines and antioxidants in fertile and subfertile female dromedaries. Transforming growth factor ά (TGFά) concentration was not different (*p* > 0.05) among groups. Contrariwise, interleukin ß (IL-ß) concentration differed (*p* < 0.05) among female groups. IL-ß concentration was high (*p* < 0.05) in the females suffering from ovarian hydro-bursa (BU; 10.83 pg/mL), cervical inflammation (CE; 11.69 pg/mL), and vaginal inflammation (VA; 8.59 pg/mL), and medium in fertile females (CON; 4.87 pg/mL), females having uterine inflammation (EN) and Fallopian tubes adhesions (SA; 3.18 and 3.25 pg/mL, respectively) and low in the case of inactive ovaries (IO; 0.26 pg/mL).

Reduced glutathione (GSH), as an antioxidant, revealed the highest (*p* < 0.05) concentration in the fertile females (491.94 µmol/L), followed by the females having cervicitis (CE; 473.04 µmol/L), but both these two groups surpassed (*p* < 0.05) the other infertile groups (Table 3). Nitrous oxide (NO) concentration revealed significant (*p* < 0.05) elevation in the control females, accounting for 2–8 folds of its counterpart levels in the infertile females. Females with ovarian abnormalities (BU and IO) have demonstrated the lowest concentrations of NO among other infertile groups (Table 3).

In control camel females (CON), correlations between various hormones, cytokines and antioxidants revealed negative relationship between inhibin and T4 (−0.94; *p* = 0.02), between E2 and P4 (−0.98; *p* = 0.005), between DHEA and T4 (−0.98; *p* = 0.003), between P4 and GSH (−0.96; *p* = 0.009), IL-ß and GSH (−0.92; *p* = 0.03), and the relationship between FSH and inhibin approached significance (−0.85; *p* = 0.065). On the contrary, positive relationships were found between FSH and GSH (0.88; *p* < 0.05), DHEA and inhibin (0.97; *p* = 0.006), P4 and IL-ß (0.88; *p* < 0.05), and E2 and GSH (0.90; *p* = 0.04).

In the females diagnosed with ovarian hydrobursitis (BU), the correlation was positively significant between inhibin and TGFά (0.88; *p* < 0.0001). The positive relationship between E2 and P4 in this group approached significance (0.43; *p* = 0.06). There was also found to be a positive relationship between total T4 and free T4 (0.81; *p* = 0.005). Nitrous oxide positively correlated with: FSH (0.78; *p* = 0.04), inhibin (0.78; *p* = 0.04), and TGFά (0.88; *p* = 0.01). Moreover, FSH positively correlated with free T3 (0.74; *p* = 0.01) and DHEA (0.70; *p* = 0.02). On the contrary, blood glucose negatively correlated with, FSH (−0.65; *p* = 0.04), TGFά (−0.60; *p* = 0.049), free T4 (−0.76; *p* = 0.01), and total T4 (−0.65; *p* = 0.04).

In the females with endometritis (EN), there were found positive relationships between FSH and the following: inhibin (0.45; *p* = 0.02), F-T4 (0.66; *p* = 0.028), F-T3 (0.60; *p* < 0.05), and TGFα (0.55; *p* = 0.003). Moreover, a strong positive relationship was found between inhibin and TGFα (0.86; *p* < 0.0001). In this group of subfertility, progesterone correlated positively with estradiol (0.71; *p* < 0.0001). Moreover, there exists a positive significant relationship between F-T3 and F-T4 (0.84; *p* = 0.001). Nitrous oxide approached a significant positive correlation with interleukin β (IL-β) (0.73; *p* < 0.1) in these females. It was unlikely that there were found significant negative relationships between P4 and T4 (−0.72; *p* = 0.01), and between E2 and T4 (−0.86; *p* = 0.0008). Inhibin correlated negatively with GSH (−0.64; *p* < 0.05).

Females with inactive ovaries (IO) revealed an array of positive relationships between the tested metabolites. One of the most important abnormal relationships is the positive correlation between FSH on one side and inhibin (0.52; *p* = 0.02) and TGFα (0.57; *p* = 0.009) on the other side. Moreover, a strong positive correlation was determined between inhibin and TGFα (0.93; *p* < 0.0001). Similarly, significant positive correlation coefficients were indicated between E2 and P4 (0.47; *p* = 0.04) and between E2 and NO (0.96; *p* = 0.0006). Free T4 positively correlated with both total T4 (0.90; *p* = 0.0002) and free T3 (0.60; *p* < 0.05). Meanwhile, blood glucose correlated positively with free T3 (0.61; *p* < 0.05). In terms of antioxidants, GSH positively correlated with both total T4 (0.61; *p* < 0.05) and free T3 (0.65; *p* < 0.05). Reversibly, E2 correlated negatively with T4 (−0.59; *p* < 0.05). DHEA negatively correlated with both IL-β (−0.89; *p* < 0.05) and NO (−75; *p* < 0.05).

Females having cervicitis (CE) exhibited strong positive relationships between the following: E2 and P4 (0.99; *p* = 0.009), TGFα and free T3 (0.98; *p* = 0.03), glucose and P4 (0.98; *p* = 0.02), glucose and E2 (0.94; *p* < 0.05), and inhibin and F-T3 (0.97; *p* = 0.03). However, strong negative relationships were found between GSH and P4 (−0.97; *p* = 0.03), GSH and E2 (−0.96; *p* = 0.04) and GSH and glucose (−0.96; *p* = 0.04). T4 approached significance (*p* < 0.1) in its positive relationship with free T4 (r = 0.94); meanwhile, IL-β approached significance (*p* < 0.1) in its negative relationship with DHEA (r = −0.93).

In females that suffered from vaginitis (VA), there were found significant positive relationships between inhibin and TGFα (0.76; *p* < 0.05), IL-β and TGF-α (0.79; *p* < 0.05), free T3 and free T4 (0.77; *p* < 0.05), DHEA and free T3 (0.75; *p* < 0.05) and between E2 and P4 (0.88; *p* = 0.009).

Females having adhesions in the Fallopian tubes (salpingitis; SA) revealed positive relationships between inhibin and TGFα (0.97; *p* = 0.03), inhibin and glucose (0.99; *p* = 0.007), and TGFα and glucose (0.98; *p* = 0.02). However, these females revealed significant negative relationships between P4 and inhibin (−0.94; *p* < 0.05), P4 and TGFα (−0.97; *p* = 0.03), P4 and glucose (−0.94; *p* < 0.05), and NO and free T4 (−0.98; *p* = 0.02).

## 4. Discussion

### 4.1. Impact of Heat Stress and Lack of Nutrients on Animal Reproduction

Reproductive functions in sheep are negatively influenced by ambient heat stress. As the ambient temperature elevates, the animal dissipates the excess temperature through sweating and panting [22]. The mechanism by which heat stress impairs reproductive function was due to less feed intake and higher water consumption. These two factors apparently might impair the metabolic processes, leading to the impairment of the reproductive process in dairy cows [23]. Heat stress damages the estrous cycle regularity and ovarian folliculogenesis in cattle [24] and follicular recruitment and steroidogenesis in goats [25]. Most of the infertile female camels in this study were admitted to the university hospital by poor animal raisers. These farmers usually leave their animals grazing on the rangeland’s poor natural pasture. In addition, they malpractice their animals, i.e., after mating their females, they put hot peppers in the vagina of the females believing that this practice stimulates the contractility of the genital tract to withdraw the semen internally and increase conception. About 82% of the infertile females in the current study were diagnosed with problems inherent to the ovaries and uterus. Lack of nutrients in the desert soil and thus in the herbs and plants that these animals graze on might be an essential candidate affecting its productive and reproductive efficiency. Recently, a strong reciprocal relationship was reported between estradiol and progesterone on the one side and the trace elements; Zn, Se, Mn, and Fe on the other side, of ewes and goats raised under hot climates in the Qassim region [26]. Moreover, Zaher and his coworkers found fewer copper, calcium, and phosphorus levels in the suboptimal reproductive female dromedaries [27]. Moreover, energy balance is vital in the ovarian dynamics and estrous cycle exhibition in dairy cows [28].

### 4.2. FSH, Inhibin, Thyroid Hormones, and Glucose as Bio-Indicators for Reproductive Failure

The hormonal profiles in fertile females expressed an inverse relationship between FSH and inhibin, indicating a normal hypothalamic–pituitary–ovarian axis in women [29], cattle [30], mares [31], sheep [32], and goats [33]. It has been well known for a long time that immunization against inhibin elicits the increase in FSH and estradiol secretion in cows [34] and sheep [35]. Inhibin plays a pivotal role in the folliculogenesis in the ovaries and regulates the FSH in the anterior pituitary [6]. The present study found a strong negative association between peripheral FSH and inhibin in fertile females. This relationship was reversed in the infertile females, particularly these females suffering from endometritis and inactive ovaries. This abnormal positive relationship of FSH/inhibin would result from the imbalances of metabolic hormones (i.e., free T3, total T4, and free T4) and the lower blood glucose in all infertile compared to fertile females. The sub-optimal glucose levels, especially in the females having ovarian hydro bursitis, revealed negative associations with T4 (free and total) and FSH. It has been demonstrated that the infusion of sheep ewes with glucose during the late luteal phase of the estrous cycle increased ovulation rate, FSH secretion, and metabolic hormones [36]. Thyroid hormones (free T3, free T4, and T4) have not demonstrated variations among fertile and infertile females. The thyroid hormones do not respond efficiently to the lack of feed energy, such as insulin and insulin-like growth factor-I. It has been demonstrated that T3 and T4 in the presence of normal levels of FSH and estrogen have direct stimulatory effects on ovarian function. Thyroid hormones per se cannot play vital roles in the reproductive process; instead, they work in a symphony of other hormones [37].

### 4.3. Expression of TGFά and IL-ß as a Response to the Reproductive Failure

The decreased glucose in underfed infertile females induced the liberation of higher levels of TGFά. As a proinflammatory cytokine, TGFά was a stimulator of vasodilatation and edema accumulation and contributed to oxidative stress at the location of inflammation [38]. The TGF superfamily was mentioned to regulate ovarian folliculogenesis and oocyte maturation [39]. IL-ß revealed the highest levels in the cases of ovarian hydrobursitis, cervicitis, and vaginitis. As a member of the interleukins family, IL-ß levels were high in inflammation postpartum [40]. Various research studies were reported on the expression of IL-ß in human diseases [41,42] and cow endometritis [43]. Moreover, the inflammation of the buffalo udder (i.e., clinical and subclinical mastitis) promoted the increase in cytokines (TNF-α, IL-6, IL-1β, and IFN-γ) production, which impaired the reproductive success in this animal [44].

### 4.4. Sex Steroid Hormones Response to Reproductive Failure

Ovarian steroids (estradiol and progesterone) correlated (r = −0.98) negatively in normally cycling fertile females; however, this relationship was reversed in most infertile cases. These opposite correlations in infertile females are probably ascribed to the higher production of inhibin in conjunction with the lower FSH secretion in these animals. The inverse relationship between estradiol and progesterone during regular estrous cycles was reported. The existence of either a cystic follicle or a persistent corpus luteum in ovaries, particularly in the cases of ovarian hydrobursa, inactive ovaries, endometritis, and vaginitis, might be attributed to an imbalance that the under-nutrition has caused on the hypothalamic–pituitary–ovarian axis [45]. In addition, in their study Dawuda and his associates found an impairment in the hypothalamic–pituitary–ovarian axis of beef heifers exposed to an under-nutrition program, which led to insufficient secretion of gonadotrophin and sex steroid hormones and irregularities of the ovarian kinematics [46]. DHEA did not differ among groups, even though it exhibited a strong positive relationship (r = +0.97) with inhibin in fertile females; however, it shows a robust negative relationship (r = −0.98) with total T4. Yilmaz and his coworkers demonstrated an elevation of inhibin as a response to DHEA supplementation in the diminished ovarian reserve of females [47].

### 4.5. Antioxidants’ Response to the Reproductive Inefficiency

Antioxidants as stress biomarkers revealed 2–8 folds of NO increase in fertile compared with infertile females. Moreover, GSH concentration revealed a higher value in fertile versus other infertile females. GSH is the most abundant endogenous antioxidant that protects the body’s cells and immunity [11]. The highest levels of GSH in the ordinarily fertile female camels fed optimal diets reflect the concept that general health is a function of nutritional status. Studies on various animal species demonstrated that adequate provision of protein in the diet can maintain GSH homeostasis. Thus, GSH plays a crucial role in nutrient metabolism, antioxidant defense system, and regulation of cellular functions. One of the current study findings demonstrated a strong positive relationship (r = 0.88) between FSH and GSH in the ordinarily fertile females. This relationship was weak and non-significant in all categories of infertile females. GSH deficiency is probably a contributor to oxidative stress [48]. Besides, the extreme heat burden during summer in the desert imposes less appetite, decreases weight gain, and deteriorates the reproductive cycle and ovarian functions [49]. Other studies on various animal species reported decreases in endogenous antioxidants such as GSH, vitamins A, C, and E, and β carotene [50,51]. GSH was inversely correlated with progesterone and IL-β in typically fertile females. Since the fertile females were all pregnant, the high blood progesterone could have an anti-oxidative role during gestation. This concept was recently suggested by Hernández-Rabaza and his associates on retinal diseases [52]. In postmenopausal women, GSH decreased with FSH increase [53].

Nitrous oxide was found to increase with the development of ovarian follicles and the increase in estradiol, as was found in the current study [54]. Moreover, the finding of the positive relationship between IL-β and NO in the endometritis-females was confirmed previously by [55]. They reported that IL-β is a critical component that regulates NO’s synthesis.

## 5. Conclusions

The ultimate findings of the current study demonstrate that the exposure of female dromedaries to heat stress in conjunction with a lack of sufficient nutrients and the unsanitary measures conducted by the majority of the camel owners in the desert led to hormonal imbalances, cytokine production, and reduced antioxidants. The best bio-indicator for the subfertility in female camels was blood NO. Therefore, these ambient factors impair animal reproduction and change the blood bio-indictors. To overcome these challenges, the animal raisers must abstain from implementing conventional malpractices and provide good quality concentrates andbalanced salt licks, in addition to alfalfa hay in summer or green fodder in winter. Moreover, the animals must be housed under humid-aerated shades, at least during the hottest time of the day. Further studies are warranted using a large population of fertile and impaired fertility animals.

## Figures and Tables

**Table 1 animals-12-02125-t001:** Effect of the fertility status of the female dromedary camel on reproductive hormones as FSH, inhibin, estradiol-17ß, progesterone, and DHEA* (Mean ± SEM).

			Reproductive	Hormone		
Fertility Status	No. Animals	FSH (mIU/mL)	Inhibin (pg/mL)	Estradiol (pg/mL)	Progesterone (ng/mL)	DHEA (ng/mL)
Control Fertile Females	5	3.38 ± 0.21 ^a^	35.98 ± 3.07 ^a^	603.56 ± 53.21 ^a^	2.74 ± 0.47	44.48 ± 2.36
Ovarian Hydro-bursa	19	2.17 ± 0.07 ^cd^	58.84 ± 2.66 ^b^	64.4 ± 24.38 ^b^	1.36 ± 0.28	43.71 ± 3.02
Inactive Ovaries	20	2.24 ± 0.08 ^cd^	61.79 ± 2.63 ^b^	127.56 ± 32.22 ^b^	1.48 ± 0.34	46.7 ± 2.16
Uterine Inflammation	28	2.13 ± 0.08 ^cd^	58.27 ± 2.23 ^b^	144.48 ± 38.57 ^b^	2.41 ± 0.38	49.5 ± 2.41
Cervicitis	4	2.79 ± 0.19 ^b^	68.21 ± 7.11 ^b^	215.06 ± 185.17 ^b^	0.94 ± 0.86	53.03 ± 1.55
Vaginitis	7	2.5 ± 0.23 ^bc^	66.4 ± 4.04 ^b^	218.71 ± 91.19 ^b^	2.28 ± 1.0	52.77 ± 2.61
Salpingitis	4	2.01 ± 0.05 ^d^	55.83 ± 3.95 ^b^	629.15 ± 145.22 ^a^	1.42 ± 0.77	40.76 ± 6.75

* Dehydroepiandrosterone. Means in the same column with different superscripts significantly differ at *p* < 0.05.

**Table 2 animals-12-02125-t002:** Effect of the fertility status of the female dromedary camel on metabolic hormones and glucose (Mean ± SEM).

			Metabolic	Hormone	
Fertility Status	No. Animals	Free T3 * (pg/mL)	Free T4 * (ng/mL)	T4 * (µg/mL)	Glucose * (mmol/L)
Control Fertile Females	5	3.39 ± 0.24	1.47 ± 0.08	9.14 ± 0.33	32.84 ± 2.65
Ovarian Hydro-bursa	19	3.69 ± 0.41	1.46 ± 0.09	9.67 ± 0.18	26.68 ± 2.53
Inactive Ovaries	20	3.27 ± 0.29	1.22 ± 0.1	8.99 ± 0.19	24.61 ± 2.97
Uterine Inflammation	28	2.97 ± 0.36	1.37 ± 0.06	8.92 ± 0.32	25.5 ± 1.83
Cervicitis	4	2.92 ± o.13	1.22 ± 0.2	9.57 ± 0.56	27.66 ± 3.79
Vaginitis	7	3.37 ± 0.25	1.28 ± 0.11	9.03 ± 0.19	24.42 ± 3.41
Salpingitis	4	3.11 ± 0.63	1.31 ± 0.1	9.43 ± 0.2	18.33 ± 2.33

** p* > 0.05.

**Table 3 animals-12-02125-t003:** Effect of the fertility status of the female dromedary camel on cytokines and antioxidants (Mean ± SEM).

	No. Animals		Cytokine		Antioxidant
Fertility Status		TGFά (pg/mL)	IL-ß (pg/mL)	GSH (µmol/L)	NO (µmol/L)
Control Fertile	5	13.32 ± 1.0	4.87 ± 0.62 ^b^	491.94 ± 44.2 ^a^	156.98 ± 25.08 ^a^
Ovarian Hydro-bursa	19	12.81 ± 0.56	10.83 ± 3.78 ^a^	356.89 ± 43.24 ^bc^	19.76 ± 4.6 ^b^
Inactive Ovaries	20	12.93 ± 0.48	0.26 ± 0.07 ^c^	289.88 ± 35.77 ^c^	26.71 ± 16.53 ^b^
Uterine Inflammation	28	12,78 ± 0.38	3.19 ± 1.51 ^b^	288.68 ± 39.6 ^c^	22.37 ± 8.92 ^b^
Cervicitis	4	15.17 ± 1.17	11.69 ± 0.87 ^a^	473.04 ± 47.21 ^ab^	54.63 ± 32.14 ^b^
Vaginitis	7	14.81 ± 1.0	8.59 ± 0.71 ^a^	333.4 ± 26.4 ^c^	74.03 ± 36.73 ^b^
Salpingitis	4	12.36 ± 0.72	3.25 ± 0.91 ^b^	322.27 ± 33.66 ^c^	78.01 ± 51.82 ^b^

Means in the same column with different superscripts significantly differ at *p* < 0.05.

## Data Availability

None of the data were deposited in an official repository. The data that support the findings of this study are available from the corresponding author upon reasonable request.

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
