# Peer review of "Expression of Hormones, Cytokines, and Antioxidants in Heat-Stressed Subfertile Female Dromedaries"

_animals, 2022, doi:10.3390/ani12162125_

Round 1

Reviewer 1 Report

Research Question:

I am unclear about the link the authors are trying to make between Endometritis and Heat Stress?  Endometritis is an infection of the uterus and caused by pathogenic agents such as chlamydia rather than directly as a result of Heat Stress. Can the authors remove Heat Stress from the manuscript as there is no objective measurements taken to score HS and there is no justification provided whether the equine patients had suffered HS. 

Ethics approval: Can you provide the Ethics approval number please.

Methods

I am unsure why the inter- assay CV% is not reported for several of the assays? 

Could the assay results be made available in the supplementary.

Author Response

REVIEWER # 1

  • At line 359 and 360, a statement was added to clarify the impact of unsanitary measures that camel breeders practice on the uterine hygiene.
  • The word “Expression” in the title doesn’t solely refers to the “GENE EXPRESSION”, rather it refers to “a transforming state of what is expressed under certain circumstance”.
  • Ethical approval contact number has been added at line 371.
  • The inter-assay CVs for each parameter were added.
  • The assay parameters excel sheet has been provided as a supplementary.

Reviewer 2 Report

The manuscript “Expression of hormones, cytokines, and antioxidants in heat-stressed subfertile female dromedaries” aimed at exploring blood metabolites such as metabolic and reproductive hormones, cytokines, and antioxidants in subfertile female camels. The manuscript is interesting and some comments/suggestions, I enumerate below:

In the abstract:

1. In the phase “this study aimed to investigate the expression of inhibin, TGFά, ILß, FSH, sex and metabolic hormones, and antioxidants for the fertility status in camel females”, I think that the term "expression" would not be correct, since the authors quantified these components, but did not evaluate gene expression itself.

2. Define “DHEA”.

In introduction:

1. I suggest that the authors present more works that justify the proposal of the manuscript. The authors bring data on these components in folliculogenesis in some species, but I think it is also interesting to address what information there is in dromedaries and what justifies investigating through the generation of the scientific hypothesis.

In the material and methods:

1. The difference between fertile and subfertile animals seems big to me. How could the authors explain that this difference did not affect the results obtained?

2. For all blood tests performed, I suggest entering how many samples were taken and in what period of time.

3. For all blood tests performed, I suggest referencing.

In the results and discussion:

1. In the result “Even though the mathematical value of progesterone was highest in control females, the differences among groups were not significant (P >0.05)”, I suggest the authors re-write the sentence as it either gives a statistical difference or it doesn't give a statistical difference. A mathematical value is not interesting in writing results.

2. In the “Impact of heat stress and lack of nutrients on animal reproduction”, I cannot see in this item a discussion of the results, but an introduction to the manuscript. I suggest that the authors rewrite so that indeed the correlations from the literature exist with the results.

3. In the “Studies on humans and animals”, I suggest the authors specify the species.

4. I suggest that the authors better address the importance of finding biomarkers of subfertility in dromedaries and this should also be presented more adequately in the introduction.

Author Response

REVIEWER #2

  • Again the word “expression” doesn’t only mean “gene expression” but in its wide meaning refers to a response of the cells to a specific circumstance.
  • DHEA has been defined at line 53.
  • A statement showing the reproductive traits in the one-humped camel is added at line 86-90 (REF 9).
  • Generally, the animals submitted to the university hospital are those suffering from reproductive failure, however the fertile females that regularly admitted to the hospital are owned by few number of the owners who don’t owe fertile bulls. Thus they brought their fertile females to be AI under the Vets supervision. To avoid the differences in sampling numbers, the data were analyzed by ANOVA with unequal subclass numbers.
  • Only one sample has been taken out of each female at its entrance to the clinic combined with the hand and ultrasound examination. LINE 103.
  • References are inserted for all tested blood parameters. LINES 109, 112, 118, 119, 122, 131,& 135.
  • The sentence (Even though the mathematical value of progesterone was highest in control females, the differences among groups were not significant (P >0.05)) has been rewritten. LINE 184.
  • The first sentence under title” Impact of heat stress and lack of nutrients on animal reproduction” has been deleted. LINE 267.
  • The sequence of the sentence regarding biomarkers in subfertility has been properly made in the INTRODUCTION. LINE 79.
  • All added references in the text have been mentioned in the REF LIST.